# Characterization, Biocompatibility, and Optimization of Electrospun SF/PCL/CS Composite Nanofibers

**DOI:** 10.3390/polym12071439

**Published:** 2020-06-27

**Authors:** Hua-Wei Chen, Min-Feng Lin

**Affiliations:** Department of Chemical and Materials Engineering, National Ilan University, Yilan 26047, Taiwan; superlemi@gmail.com

**Keywords:** chitosan, composite nanofibers, electrospinning, silk fibroin, polycaprolactone, Taguchi

## Abstract

In this study, composite nanofibers (SF/PCL/CS) for the application of dressings were prepared with silk fibroin (SF), polycaprolactone (PCL), and chitosan (CS) by electrospinning techniques, and the effect of the fiber diameter was investigated using the three-stage Taguchi experimental design method (L9). Nanofibrous scaffolds were characterized by the combined techniques of scanning electron microscopy (SEM) and transmission electron microscopy (TEM), a cytotoxicity test, proliferation tests, the antimicrobial activity, and the equilibrium water content. A signal-to-noise ratio (S/N) analysis indicated that the contribution followed the order of SF to PCL > flow rate > applied voltage > CS addition, possibly owing to the viscosity and formation of the beaded fiber. The optimum combination for obtaining the smallest fiber diameter (170 nm) with a smooth and uniform distribution was determined to be a ratio of SF to PCL of 1:2, a flow rate of 0.3 mL/hr, and an applied voltage of 25 kV at a needle tip-to-collector distance of 15 cm (position). The viability of these mouse fibroblast L929 cell cultures exceeded 50% within 24 hours, therefore SF/PCL/CS could be considered non-toxic according to the standards. The results proposed that the hydrophilic structure of SF/PCL/CS not only revealed a highly interconnected porous construction but also that it could help cells promote the exchange of nutrients and oxygen. The SF/PCL/CS scaffold showed a high interconnectivity between pores and porosity and water uptake abilities able to provide good conditions for cell infiltration and proliferation. The results from this study suggested that SF/PCL/CS could be suitable for skin tissue engineering.

## 1. Introduction

In recent decades, nanofiber membranes have been widely used in various fields and attracted more attention due to their unique properties, such as large specific surface areas, high porosity, interconnected pores, and high functionality. Electrospun composite nanofibers possess great potentialities in biomedical applications, such as tissue engineering [2,3,4], wound healing [5,6,7], and drug delivery [8,9,10], as well as other applications such as magnetism [11], photonics [12], filtration [13], composites [14], shape memory [15], and lithium batteries [16]. The diameter of electrospun nanofibers affects many important properties, such as the melting point, tensile modulus, hardness, drug delivery, biological factors, and cell growth of nonwoven fabrics [17,18]. Research [19] has demonstrated that fiber diameter plays a key role in cell adhesion, proliferation, and cell migration on the scaffold.

Silk fibroin (SF), which is extracted mainly from silkworms, has various properties, including good biocompatibility, biodegradability, morphological flexibility, mechanical properties, a low inflammatory response, non-toxicity, and non-carcinogenicity, and it can promote cell adhesion, migration, and the proliferation of cell ligands [8,20,21]. However, the β-sheet secondary structure of pure SF seems to impede the electrospinning process, and the mechanical properties of neat SF electrospun fibers were poor [20]. Herein, polycaprolactone (PCL), the most commonly used synthetic polymer, was chosen to be blended with SF. PCL is widely used in tissue engineering and drug delivery applications due to its good mechanical properties and biodegradability. PCL has limitations to its biological activity, hydrophobicity, and bacterial degradation, therefore PCL cannot provide the adhesion environment required for cells [2]. PCL has renowned mechanical properties and does not have ionizable side groups in its structures, such as –COOH and –NH_2_, which occur on the natural polymer chitosan (CS) and on several anionic polysaccharides and proteins, respectively. The CS is an amino polysaccharide derived from chitin that has excellent biological properties, such as biocompatibility, biodegradability, hydrophilicity, non-toxicity, and antithrombotic and antimicrobial activities. Chitosan that possesses positively charged groups (amine groups) is likely to interact with negatively charged cell membranes via electrostatic interaction. Furthermore, the antimicrobial activity of chitosan effectively increases the permeability of negatively charged cell membranes to disrupt and release intracellular compounds [22]. Therefore, investigations of SF and its association with other components (PCL and CS) are carried out to give better mechanical properties to the smooth nanofibers for the tissue engineering. The Taguchi method is an effective method to find the influence of different factors on the target results, thereby improving the manufacturability, reliability, and quality of a product and reducing the number of experiments and calculation time [23,24]

Thus, SF/PCL/CS composite nanofiber scaffolds for the application of dressings and tissue engineering were fabricated with PCL polymer as a precursor using electrospinning techniques in this study. Further, the effect of chitosan additions on the nanofiber diameter was also investigated via the analysis of the antimicrobial activity and equilibrium water content. The optimum combination of parameters obtained from the ratio of silk fibroin to polycaprolactone, chitosan additions, flow rate, and applied voltage in response to minimizing diameter size and its variation for SF/PCL/CS composite nanofibers was determined by means of the Taguchi DoE method. Herein, the biocompatible properties were evaluated with cytotoxicity tests and proliferation tests so as to determine the optimal SF/PCL/CS scaffolds.

## 2. Experimental Method

### 2.1. Preparation of Regenerated Silk Fibroin (SF)

All the materials, solvents, and reagents were purchased from commercial suppliers and used as received. Cocoons were received from Paolun farm (Taiwan). Polycaprolactone (PCL, Mw = 80,000) was purchased from Sigma-Aldrich (St. Louis, MO). Polyethylene oxide (PEO, Mw = 60,000–100,000), chitosan (CS, Mw = 10,000–30,000), and formic acid were purchased from Acros Organics. The preparation of the silk fibroin (SF) used in this study made reference to previous research [1], with some modification. The cocoons were boiled in a 0.5% (w/v) Na_2_CO_3_ aqueous solution at a temperature of 100 °C for one hour. The silk fibers were rinsed with distilled water for 30 min to remove the Na_2_CO_3_ aqueous solution and then rinsed with deionized water for 30 min to remove the sericin. After being dried to a constant weight in an oven at 80 °C, the degummed silk fibers were dissolved in 40% aqueous CaCl_2_ at 100 °C. The SF solution was replaced with a dialyzed membrane (molecular weight cut-off (MWCO) = 12,000−14,000 Da) for three days to eliminate small molecular impurities and calcium chloride. Lastly, the SF solution was lyophilized in a freeze dryer and stored at room temperature.

### 2.2. Preparation of the Electrospinning Solutions and Electrospinning

The SF and PCL were dissolved in formic acid to obtain 10 wt.% concentrations. Different ratios of SF/PCL (5.00%:5.00%, 3.33%:6.67%, and 2.50%:7.50%) were dissolved in formic acid and stirred at room temperature for two hours. Subsequently, 1 wt.% PEO as a thickener was added to the solution. Finally, from 0.5 to 1 wt.% CS was added to the electrospinning solutions.

The SF/PCL/CS composite nanofibers were obtained from the electrospinning of the prepared suspensions through a FES-COS electrospinning apparatus (Falco Co, Taipei, Taiwan). Briefly, the suspensions were drawn into a 10 mL syringe with a 21-gauge needle. The electrospinning was performed under ambient conditions (a temperature of 24.5 to 27.5 °C and a relative humidity of 45% to 50%). The following optimized electrospinning parameters were kept constant throughout the experiments: 100 rpm roller collector to collect fibers, 15 cm TCD (tip to collector distance), 15 kV–25 kV applied voltage, and 0.2 mL/h–0.4 mL/h feeding rate. The SF/PCL/CS homogeneous solutions were electrospun on a rotating cylindrical drum covered with an aluminum layer during the process. Finally, the collected membranes were taken from the surface of the collector and conserved in a sealed container for further experiments.

### 2.3. Taguchi DOE Parameter Setting

Numerous references [19,20,21,22,23,24] state that the ratio of silk fibroin to polycaprolactone, the chitosan content, the flow rate, and the applied voltage have significant effects on the average diameter and uniformity of fibers; thus, these four concentration and electrospinning parameters were selected for this experiment (Table 1).

The full factorial experiment of 81 (3^4^) trials could be completed in just 27 runs due to the slope collector; however, that would entail a large number of tests, which would be significant in both experimental cost and time. As a result, the Taguchi design of experiments (DoE) layouts were more applicable when compared to a traditional full-factorial counterpart because they reduced the number of tests to a practical level. The L9 DoE orthogonal array was selected with the assumption of no factorial interactions, resulting in nine trials, as illustrated in Table 2.

In the “larger the better” characteristic, the formula for calculating the ratio of S/N as the best parameter for calculating the factors was calculated by the following Equation (1):(1)$$S/N=-10\;\times \;\log\;(\frac{1}{n}\sum_{i=1}^{n}\frac{1}{y_{i}^{2}})$$
where n and y denote the number of measurements and observed data, respectively.

### 2.4. Characterization of Nanofiber Scaffolds

The morphology of the nanofiber scaffold was detected using scanning electron microscopy (SEM; TS 5136MM, TESCAN, Czech Republic). The average fiber diameter was determined by measuring 100 fibers selected randomly from each sample. Chemical analysis was performed using a Fourier transform infrared spectrometer (FTIR; Spectrum 100, Perkin Elmer, USA) with a scan range of 4000 to 450 cm^−1^ and an accumulation of 16 scans.

According to standard method of Japanese Industrial Standards (JIS) 10099A, the water vapor transmission rate (WVTR) is a measure of the passage of water vapor through a substance. In addition to measurements of the permeability of the vapor barriers, the porosity of the SF/PCL/CS was also examined by the study [25]. For the antibacterial assay, the inhibitory effects of chitosan on bacterial growth were detected by the plate well diffusion method [26] via the formation of a zone of inhibition. To attain this figure, the sequential dilution was necessary (six for *E. coli* and for *S*. *aureus*) according to the simultaneous counting of plate colonies (CFU). The procedure used in this analysis followed the agar diffusion method according to the previous literature [27], in which small circular cavities were punctured in the culture medium for each chitosan concentration.

The equilibrium water content (EWC) was measured by the conventional gravimetric method. The pre-weighed dry samples were immersed in deionized water, and the excess surface water was blotted out with absorbent paper. The swelling procedure was repeated until there was no further weight increase. The EWC was calculated as the weight increase with respect to the weight of the swollen samples within 24 h using the following Equation (2):(2)$$EWC=\frac{W_{\mathrm{wet}}-W_{\mathrm{dry}}}{W_{\mathrm{wet}}}\times 100\%$$
where *W*_wet_ and *W*_dry_ denote the weights of the swollen and dry samples, respectively.

A cytotoxicity assay is a test for analyzing the cytotoxic effects of materials and medical devices on living organisms [28]. The following cell culture-based tests, as recommended by ISO 10993-5, used a direct contact test. In all the tests (blank, negative control, positive control, and sample), the incubation time of the mouse fibroblast L929 cell cultures was 24 h. Cell culture is the process by which cells from human tissue are grown in an incubator under controlled conditions in order to provide sufficient material for testing. After solubilization, the solutions were transferred to fresh flat bottom 96-well plates, and the absorbance (570 nm) of each well was measured by spectrophotometry. The background absorbance at 650 nm was subtracted from the readings at 570 nm to obtain the final optical density (OD). The following Equation (3) was used to calculate the reduction in the culture viability of the cells exposed to a tested sample (i.e., SF/PCL/CS under optimal conditions) in comparison to the cell culture viability of group b:(3)$$\mathrm{Viability}\;\left(\%\right)=\frac{OD_{570e}}{OD_{570b}}\times 100\%$$
where *OD*_570*e*_ is the average OD of the respective groups that were in contact with different lots of the product and *OD*_570*b*_ is the average OD of all the wells of group b. All the values were final ODs after the subtraction of background absorbance.

## 3. Results and Discussion

### 3.1. Optimum Combination of Factors for the Application of Dressings

#### 3.1.1. Taguchi Method to Optimize Dressings

Uniform fiber diameters and smaller diameters in the scaffold for the application of dressings can provide a higher surface area and interconnected holes to promote the exchange of nutrients and oxygen and enhance the proliferation ability of cells. In the formation of nanofibers, the ratio of silk fibroin to polycaprolactone, the chitosan content, the flow rate, and the voltage are crucial factors affecting the nanofiber diameter. In Taguchi designed experiments, the higher values of the signal-to-noise ratio (S/N) identify control factor settings that minimize the effects of the noise factors. Using the Equation (1) of the smaller the better method (i.e., the smallest diameter of the nanofibers was selected based on maximum S/N ratio), the S/N ratio of the fiber fineness (the diameter of the nanofibers) in the SEM micrograph of the nanofiber could be calculated, and the results are shown in Table 2. According to the results from the L_9_ (3^4^) sample, the average S/N ratio of the four factor levels was calculated to perform the next analysis, as listed in Table 3. The influence of the four factors on the fiber diameter followed the order of the ratio of silk fibroin and polycaprolactone (∆ = 1.80) > flow rate (∆ = 1.46) > applied voltage (∆ = 1.26) > chitosan addition (∆ = 1.10). This finding suggested that the ratio of silk fibroin and polycaprolactone was the most significant factor for achieving a small electrospun nanofiber diameter for the application of dressings. Table 3 also shows the contribution of the four parameters to the influence of the SF/PCL/CS composite nanofiber diameter. The ratio of the silk fibroin and polycaprolactone was an important factor affecting the diameter of the nanofibers due to the highest contribution percentage (32.0%). Furthermore, the results indicated that the ratio of silk fibroin and polycaprolactone affected the viscosity of the electrospinning solution to produce a stable Taylor cone. The results were in accordance with the Taguchi experimental S/N design.

The average value of the confirmation experiments was 170.00±55.76 nm under the optimal parameters of a silk fibroin to polycaprolactone ratio of 1:2, a chitosan addition of 0.5%, a rate of advancement of 0.3 mL/h, and an operating voltage of 25 kV (Table 2), which provided the highest average value compared with the nine groups of quality data. As shown in Figure 1 and Figure 2, the results indicated that the optimal parameters inferred by the Taguchi method had a smaller fiber diameter, were smoother, and had a low distribution. Conclusively, the comparison of the optimal parameters inferred by the Taguchi method with the data results of the orthogonal table proved that the inferred optimal parameters were appropriate.

#### 3.1.2. Porosity and Water Uptake Abilities

The adequate pore size and interconnected pores of a scaffold provides a sufficient opportunity for cell migration and proliferation. The ability of a dressing to control water loss can be determined by the water vapor transmission rate (WVTR). The ability of a scaffold to preserve water is also important in order to evaluate its property for tissue engineering. Table 2 describes the WVTR of the SF/PCL/CS nanofibers. The porosity of the nine samples (L_9_ (3^4^)) of the SF/PCL/CS nanofibers in this study was more than 80% and was significantly higher than the pure PCL scaffolds, which had a 70% porosity, as reported by [29]. The highly porous SF/PCL/CS nanofibers could provide an appropriate environment for initial cell growth (by their structural stability), accelerated degradation (by their large surface area), and the sustained delivery of bioactive molecules (by their high porosity).

In this study, SF/PCL/CS nanofibers with graded WVTR were prepared by changing the porosity of the membrane (Table 2). The corresponding average WVTR of the samples was in the range of between 4330.71 and 4768.71 g m^−2^·24 h (extremely high permeability, L_9_ (3^4^)). An extremely high WVTR may lead to the dehydration of a wound, whereas an unacceptably low WVTR may cause the accumulation of wound exudates. Hence, a dressing with a suitable WVTR is required to provide a moist environment that can establish the best environment for natural healing. Thus, a dressing prepared by optimal condition with a WVTR of approximately 4768.71 g/m^2^·24 h could also maintain the optimal moisture content for the proliferation and function of cells and fibroblasts. According to SEM and the appropriate porosity of the SF/PCL/CS nanofiber, it was considered to have great potential for skin tissue engineering due to its interconnected pore network and suitable WVTR.

#### 3.1.3. Cytotoxicity Tests

Cytotoxicity assays are necessary for the assessment and characterization of the potentially toxic and harmful effects of a biomaterial’s compounds [28]. They are a feasible and reliable in vitro technique used for the biocompatibility evaluation of materials. Table 4 shows phase contrast images of the cultures in the experiment after 24 h, including blank, negative control (polyethylene, PE), positive control (dimethylsulfoxide, DMSO), and SF/PCL/CS under optimal conditions. In cultures exposed to blank, negative control, positive control, and SF/PCL/CS, the viability was 100, 100, 13, and 57% at 24 h of continued growth in the culture, respectively. A tested product (SF/PCL/CS) has non-cytotoxic potential for application to tissue engineering when the cell culture viability increases to >50% in comparison to the positive control (dimethylsulfoxide, DMSO), which was set at a 13% viability. The study also proposed the similar results that the tested product could be considered non-toxic as the viability of these cultures exceeded 50% [30]. Furthermore, the viability of 57% for the tested product (SF/PCL/CS) in this study was higher than that of 50.2% for the CNTs-doped PLGA nanofibers [31]. The research [31] show that the PLGA and the HNTs- or CNTs-doped PLGA nanofibers display appreciable MTT formazan dye sorption, corresponding to a 35.6–50.2% deviation from the real cell viability assay data. From Figure 3, the DMSO treatment substantially altered the morphology and attachment of cells in concurrence with a significant reduction in the cell viability. The results showed that the electrospun scaffolds (SF/PCL/CS) could support the attachment and the proliferation of mouse fibroblast L929 cells. In addition, the cells cultured on the scaffolds exhibited normal cell shapes. The obtained results confirmed the potential for use of the electrospun fiber as scaffolds for skin tissue engineering.

### 3.2. Effect of the Ratio of Silk Fibroin to Polycaprolactone on Fiber Diameter

In the electrospinning process, the solution concentration is considered to be the most important parameter affecting the fiber morphology [31,32]. Figure 4 shows the effect of the ratio of silk fibroin to polycaprolactone on the fiber diameter. At a silk fibroin to polycaprolactone ratio of 1.0, the entanglement between the polymer chains formed obvious beads because the low viscosity of the solution did not provide a stable jet. As the ratio of silk fibroin to polycaprolactone decreased from 0.5 to 0.25, the average fiber diameter increased from 208.27±68.27 nm to 664.23±131.54 nm in this study. The main reason was that the increase in viscosity hindered the bending stability of the jet to produce a coarser fiber [33,34].

### 3.3. Effect of Chitosan Addition on Fiber Diameter, Antimicrobial Activity, and Equilibrium Water Content

Figure 5 depicts the evaluation results of the fiber diameter of the electrospinning scaffold with different chitosan additions. The average fiber diameter decreased from 523.23±92.60 mm to 181.45±41.57 nm with the increased chitosan addition from 0.25% to 1.00% because of the charge density. The chitosan addition enhanced the higher charge density of the jet to produce the thinner fibers due to the increase in conductivity [21,35]. Compared to the chitosan addition of 1.00%, the average fiber diameter of the electrospinning scaffold was increased to 308.90±74.98 nm with a chitosan addition of 1.50%, because the increase in viscosity caused the bending instability of the jet and the accelerating solidification of the polymer jet [34,36].

The results from Table 5 revealed the mean diameter of inhibition zone for *E. coli* with the chitosan amounts of 0.25%, 0.5%, 0.75%, and 1.00% were 65, 63, 64, and 63 mm, respectively, proving the strong antibacterial property. Moreover, the samples with the chitosan amounts of 0.25%, 0.5%, 0.75%, and 1.00% exhibited the inhibition zone diameters for *S. aureus* of 58, 55, 51, and 55 mm, respectively. As illustrated in Figure 6 and Figure 7, the activity intensity could be visually determined by agar well diffusion assay testing via assessing the local inhibition. The results of the experiments conducted using different chitosan additions had an inhibitory effect on the mean diameter of the inhibition zone for two types of bacteria (*E. coli* and *S*. *aureus*). The results exhibited better inhibitory effects against gram-positive bacterium *S*. *aureus* compared to the gram-negative bacterium *E. coli*., which was in agreement with the results attained in previously published works [37,38]. The results in this study proposed that unmodified chitosan generally acts stronger on gram-negative strains than on gram-positive strains, owing to the electrostatic interaction between positively charged R–N(CH_3_)_3_^+^ sites and negatively charged microbial cell membranes.

The hydrophilicity of nanofibers (SF/PCL/CS) with different chitosan additions (0.25%, 0.50%, 0.75%, and 1.00%) was measured by gravimetric analysis and designated by the percentage equilibrium water content (EWC), as shown in Figure 8. The EWC was increased with the increasing chitosan addition due to the functional groups of –OH and –NH_2_. The equilibrium water content (EWC) of all the nanofibers (chitosan additions of 0.25%, 0.50%, 0.75%, and 1.00%) was in the range of 500% to 950% and was significantly higher than the YY0148-2006 standard for medical dressings [39] and the hydrogels for tissue engineering [40]. The copolymerization with the zwitterionic comonomer leads hydrogels with a high equilibrium water content (EWC) of up to 700% while maintaining mechanical robustness [40].

### 3.4. Effect of Flow Rate on Fiber Diameter

In the electrospinning process, the flow rate is considered to be the most important parameter affecting the fiber uniformity [31,41]. The effect of the flow rate on the fiber diameter is shown in Figure 9a. As shown in Figure 9b, uniform beadless electrospun nanofibers were prepared via a critical flow rate of 0.3 mL/h for the polymeric solution. The shape of the Taylor cone at the tip of the capillary would not be maintained if the flow of the solution through the capillary was insufficient, and the insufficient intermolecular surface tension would be unable to resist the Coulomb force to maintain a stable jet [41]. However, at a higher flow rate, the solution would be wasted without differentiating adequately into the fibers. The short evaporation time of the solvent could result in the formation of beads and increase the average fiber diameter [32]. The morphology of the SF/PCL/CS nanofibers appeared to be inhomogeneous under a higher flow rate, as depicted in Figure 9c. The results could be explained by the inharmony between the fiber formation speed and the solution feed rate.

### 3.5. Effect of Applied Voltage on Fiber Diameter

In general, the operating applied voltage must exceed the minimum threshold applied voltage before the Taylor cone appears to form ultrafine nanofibers [23]. As shown in Figure 10, the results revealed that the average fiber diameter significantly decreased from 440.69 ± 105.47 nm to 208.27 ± 55.76 nm as the applied voltage increased from 15 to 25 kV. This result could be explained by the force imbalance between the repulsive Columbic force and the contracting viscoelastic force. With the increasing voltage, the intrinsic equilibrium became difficult to restore when the fibers were collected on the metal rod accompanied by the charge transfer.

## 4. Conclusion

SF, PCL, and CS were blended in different concentrations and compositions and were evaluated for their fiber diameter to examine the optimum values for nanofibers. The S/N analysis via the Taguchi experimental design showed that the ratio of SF to PCL was the most influential parameter on the fiber diameter. A smooth and uniform distributed SF/PCL/CS nanofiber with a fiber diameter of 170.0 nm was synthesized at the optimal parameters of a 1:2 ratio of silk fibroin to polycaprolactone, a 0.5% chitosan addition, a 0.3 mL/h rate of advancement, and an operating voltage of 25 kV. The SF/PCL/CS (under optimal conditions) with a WVTR of approximately 4768.71 g/m^2^·24 h could maintain the optimal moisture content for the proliferation and function of cells and fibroblasts. The EWC of the SF/PCL/CS nanofiber was increased from 500% to 950% by increasing the chitosan additions from 0.25% to 1.00% and was significantly higher than the YY0148-2006 standard for medical dressings. The porosity of the nine samples (L_9_ (3^4^)) of SF/PCL/CS in this study was more than 80% and was significantly higher than that of the pure PCL scaffolds. According to SEM and the appropriate porosity of the SF/PCL/CS nanofiber, it was considered to have great potential for skin tissue engineering due to its interconnected pore network and suitable WVTR.

## Figures and Tables

**Figure 1 polymers-12-01439-f001:**
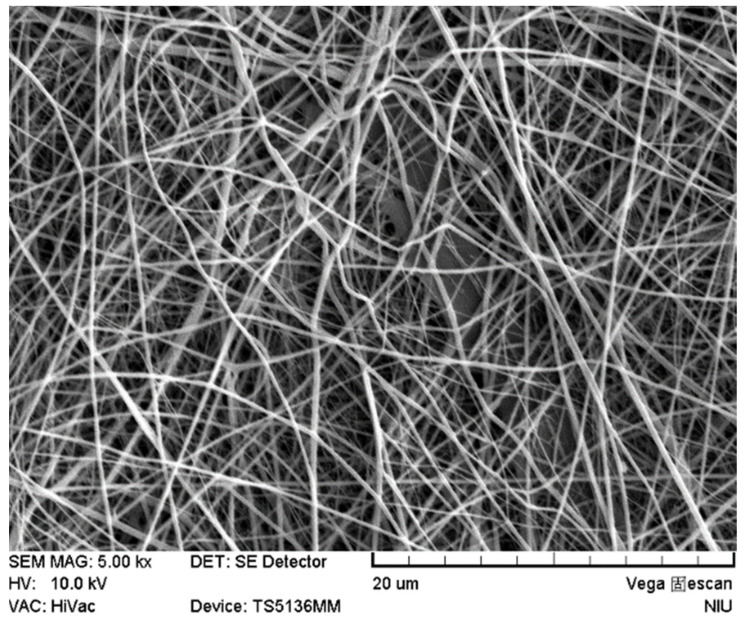
SEM photomicrograph of the silk fibroin (SF), polycaprolactone (PCL), and chitosan (CS) (SF/PCL/CS) nanofibers under optimal conditions.

**Figure 2 polymers-12-01439-f002:**
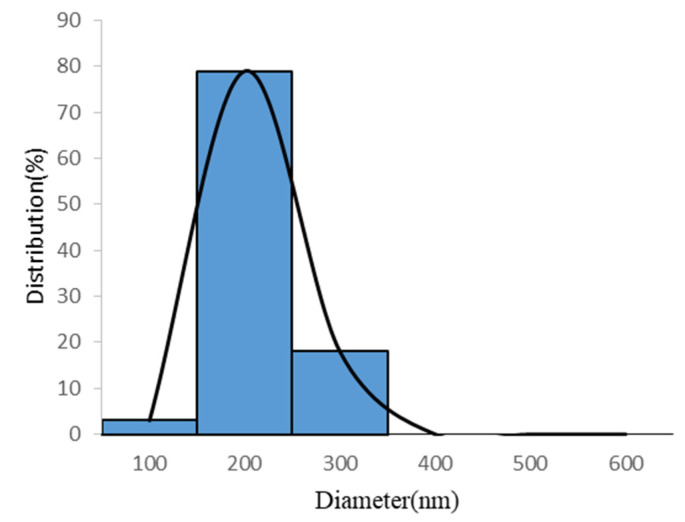
Fiber diameter distribution of the SF/PCL/CS nanofibers under optimal conditions.

**Figure 3 polymers-12-01439-f003:**
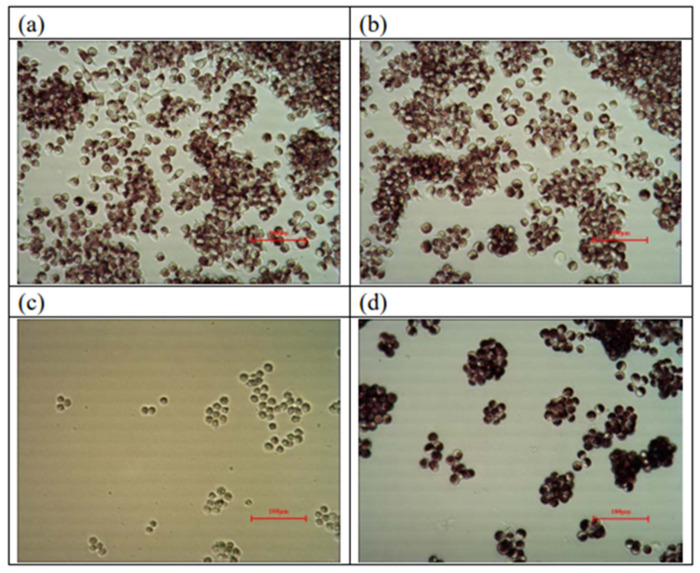
Photomicrograph of cell L 929 by direct contact method within 24 hours: (**a**) blank; (**b**) negative control (polyethylene, PE); (**c**) positive control (dimethylsulfoxide, DMSO); and (**d**) the sample (SF/PCL/CS under optimal conditions).

**Figure 4 polymers-12-01439-f004:**
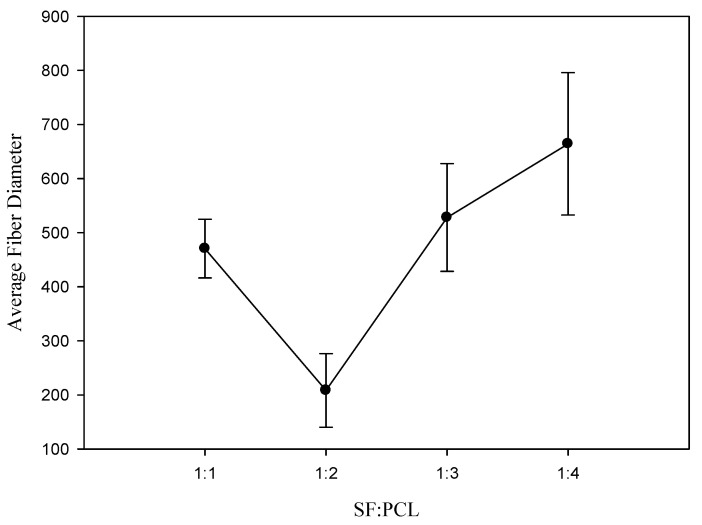
Effect of the ratio of silk fibroin to polycaprolactone on the fiber diameter.

**Figure 5 polymers-12-01439-f005:**
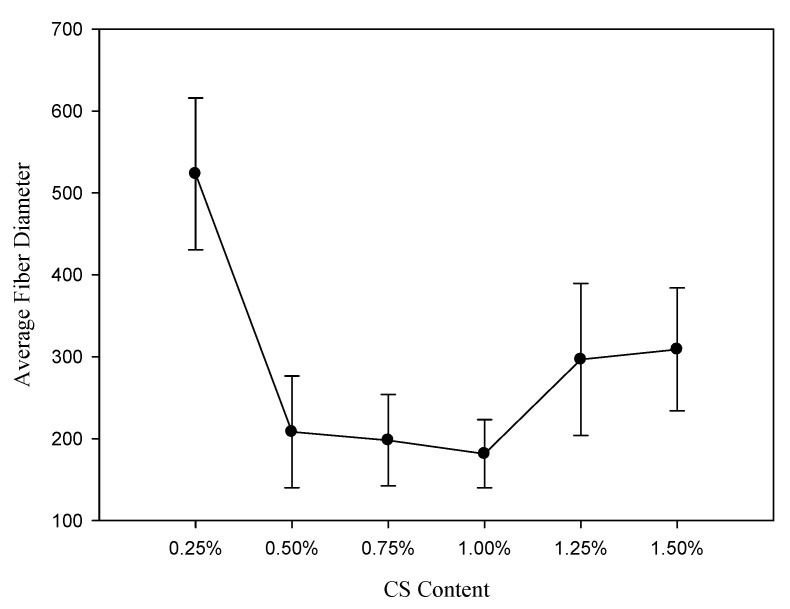
Effect of the chitosan addition on the fiber diameter.

**Figure 6 polymers-12-01439-f006:**
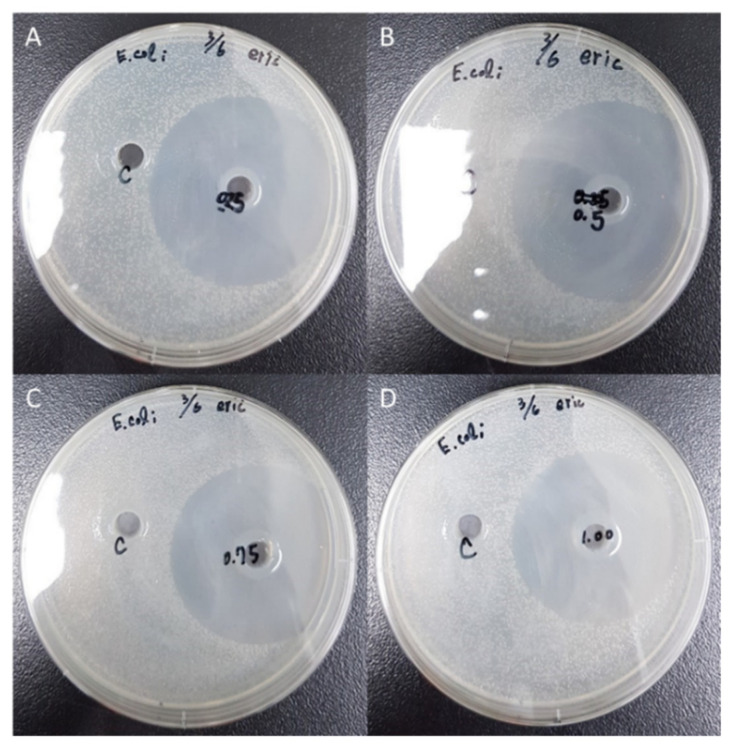
Effect of the chitosan addition on the antimicrobial activity of E. coli at different chitosan amounts: (A) 0.25; (B) 0.50; (C) 0.75; (D) 1.00 wt%.

**Figure 7 polymers-12-01439-f007:**
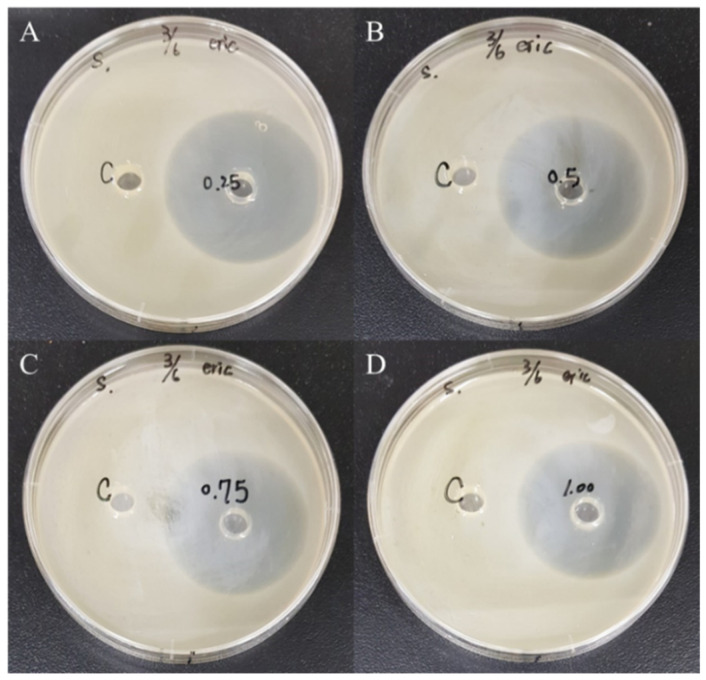
Effect of chitosan addition on the antimicrobial activity of *S*. *aureus* at different chitosan amounts: (**A**) 0.25; (**B**) 0.50; (**C**) 0.75; (**D**) 1.00 wt%.

**Figure 8 polymers-12-01439-f008:**
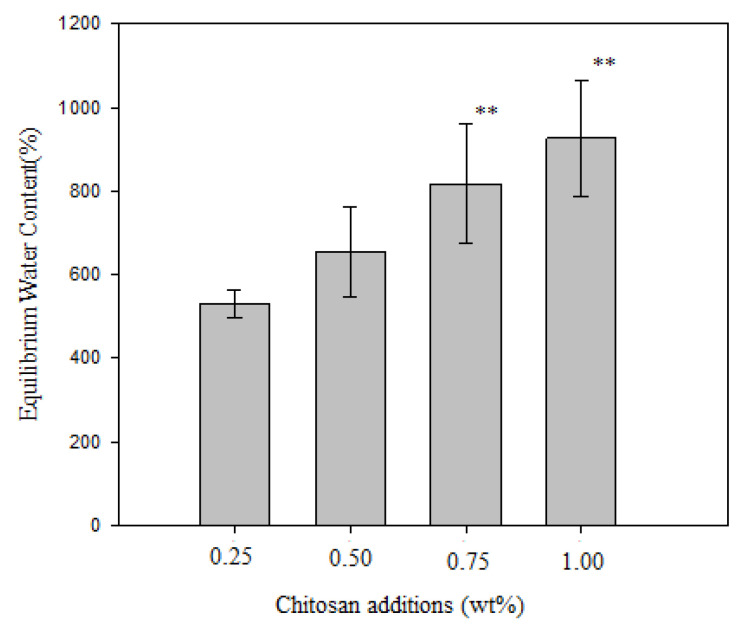
Effect of chitosan addition on the equilibrium water content.

**Figure 9 polymers-12-01439-f009:**
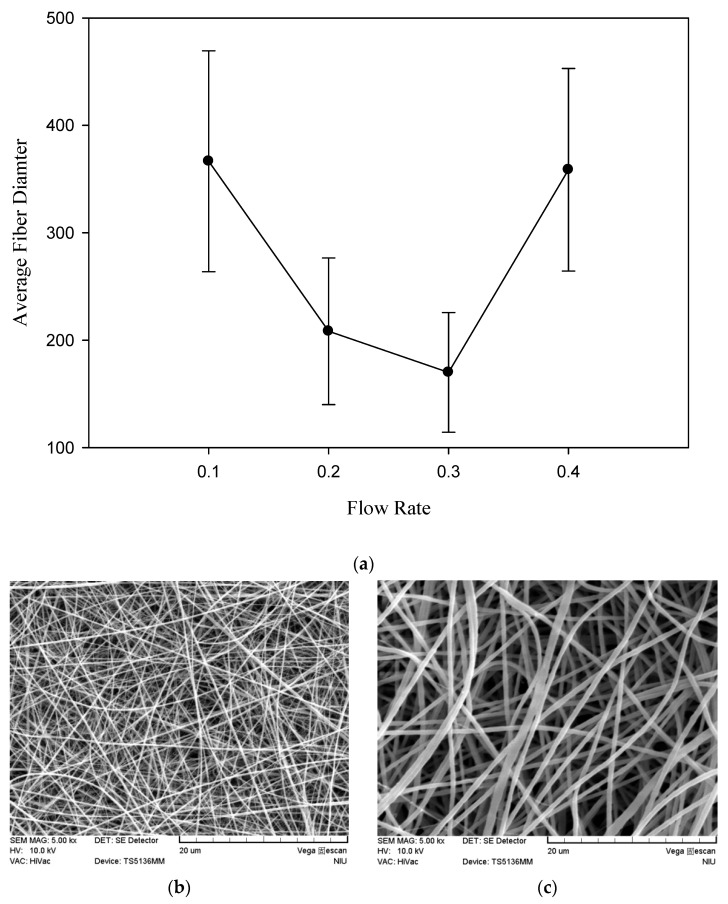
(**a**) Effect of flow rate on the fiber diameter of the SF/PCL/CS nanofibers, (**b**) SEM micrograph at 0.3 mL/h, (**c**) SEM micrograph at 0.4 mL/h.

**Figure 10 polymers-12-01439-f010:**
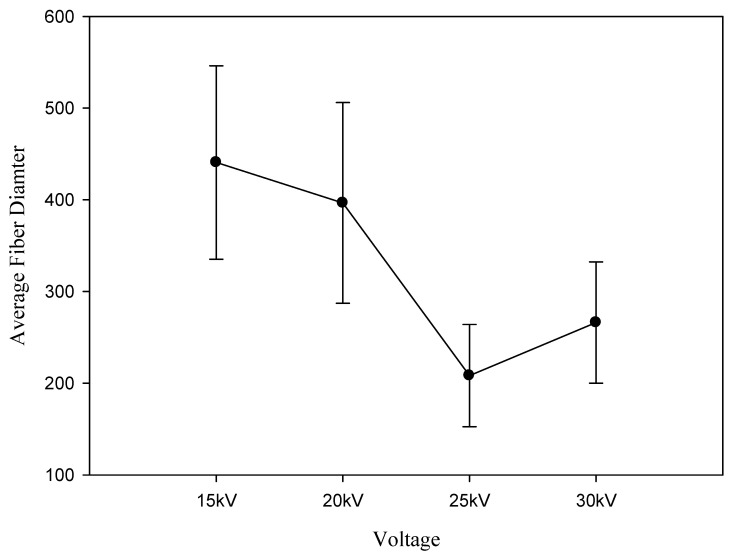
Effect of applied voltage on fiber diameter.

**Table 1 polymers-12-01439-t001:** Actors and levels used in the experiment.

	Ratio of Silk Fibroin and Polycaprolactone	Chitosan Content (wt.%)	Flow Rate (mL/h)	Voltage (kV)
**Level 1**	1:1 (5.00%:5.00%)	0.50	0.2	15
**Level 2**	1:2 (3.33%:6.67%)	0.75	0.3	20
**Level 3**	1:3 (2.50%:7.50%)	1.00	0.4	25

**Table 2 polymers-12-01439-t002:** Experimental results of the fiber fineness of the composite nanofibers planned by the L_9_(3^4^) orthogonal table.

L_9_(3^4^)	Ratio (SF: PCL)	Chitosan Addition (%)	Flow Rate (mL/h)	Voltage (kV)	S/N	Means of Diameter (nm)	Porosity (%)	WVTR (g m^−2^·24 h)
1	1:1	0.50	0.2	15	12.65	232.55±60.28	88.01±4.32	4417.29±87.27
2	1:1	0.75	0.3	20	12.71	229.09±60.66	83.10±4.21	4362.97±91.67
3	1:1	1.00	0.4	25	13.61	208.39±55.81	91.96±5.16	4641.38±19.21
4	1:2	0.50	0.3	25	15.37	170.00±55.76	92.05±3.70	4768.71±85.04
5	1:2	0.75	0.4	15	11.92	253.42±65.37	91.43±5.50	4636.29±25.46
6	1:2	1.00	0.2	20	13.11	220.67±62.13	85.37±4.04	4432.89±45.84
7	1:3	0.50	0.4	20	10.86	285.71±78.08	92.60±4.10	4581.97±92.58
8	1:3	0.75	0.2	25	11.47	266.75±76.71	84.74±5.70	4330.71±5.09
9	1:3	1.00	0.3	15	12.68	231.94±63.64	82.03±1.45	4405.41±74.86

**Table 3 polymers-12-01439-t003:** Smaller the better of the fiber fineness signal-to-noise ratio (S/N) analysis.

	Ratio of Silk Fibroin and Polycaprolactone	Chitosan Content	Flow Rate	Voltage
1	12.99	12.96	12.41	12.42
2	13.47	12.03	13.59	12.23
3	11.67	13.13	12.13	13.48
∆	1.80	1.10	1.46	1.26
Factor influence order	1	4	2	3
Contribution (%)	32.0	19.1	26.2	22.7

**Table 4 polymers-12-01439-t004:** Viability of cell L 929 according to the MTT test method.

Test Item	Absorbance (%)	Viability (%)
Blank	0.504 ± 0.011	100
Negative control	0.502 ± 0.005	100
Positive control	0.064 ± 0.002	13
Sample (SF/PCL/CS)	0.288 ± 0.020	57

**Table 5 polymers-12-01439-t005:** Effect of chitosan addition on the mean diameter of the inhibition zone of *E. coli* and *S*. *aureus* at different chitosan amounts.

Name of the Sample	Conc. (%)	Mean Diameter of Inhibition Zone (mm)	
		Test Organisms	
		Staphylococcus Aureus	Escherichia Coli
Chitosan	0.25	58 ± 4	65 ± 0
	0.50	55 ± 1	63 ± 4
	0.75	51 ± 1	64 ± 0
	1.00	55 ± 1	63 ± 2

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
