# Peer review of "Characterization, Biocompatibility, and Optimization of Electrospun SF/PCL/CS Composite Nanofibers"

_polymers, 2020, doi:10.3390/polym12071439_

Round 1

Reviewer 1 Report

The manuscript described using orthogonal experimental design to develop electrospinning nanofiber membranes made from formulas containing natural silk fibroin and biofriendly polymers and assess their capability for cell culturing and bacterial inhibition. The methods are well defined. Porosity, water uptake rate, and cytotoxicity of nanofiber membranes were studied. The effects that influence nanofiber diameter were also studied. Especially, the anti-bacterial effects with different contents of chitosan was studied against two different species of bacterial. The results are mostly supportive for what the authors are trying to claim. I would suggest accepting this manuscript to be published in Polymers if the following points could be made clearer.

  1. In the introduction, the authors didn’t elaborate clearly why they chose to make a composite fiber membrane with the selected components including silk fibroin, PCL and chitosan. It could be implied that these components have all been used individually for bio-related applications. However, the authors fail to tell the readers why they chose them and why as composites. There are already many nanofiber membranes for the proposed applications as described in this manuscript.
  2. In Table 1, is the relative ratio of chitosan and SF/PCL going to influence the results? Also, is it going to be easier for readers to follow if the ratios of each component were addressed as the percentages out of everything in total, e.g., in Level 1, chitosan takes 0.5 wt.%, SF takes 49.75 wt.%, and PCL takes 49.75 wt.% of the formulation? Or maybe the contents of chitosan are not as important from the following discussions in the manuscript since it takes only less than 1% for all the formulations.
  3. There is a spelling error in Table 4, by blanck, the authors probably meant Also, since the ratio of silk fibroin and polycaprolactone has already been defined as factor A in Line 166-167, why don’t you make it simple and use the defined term in the following texts or not defining factor A at all since it appeared only once in the whole manuscript?
  4. Starting from Line 177 to Line 179, the authors brought up the optimal parameters and conditions for fabrication. Although it is clear from previous discussion that factor A plays the most important role among all the four factors that were controlled, how each parameter in the three levels were determined was not explained. In other words, where does the conclusions about optimal parameters starting from Line 177 to Line 179 come from? Were they selected based on minimum S/N ratio? If they were selected based on the diameter of the nanofibers, under which circumstance, the smallest is believed to be the best, shouldn't the optimal ratio between SF and PLC 1:2 rather than 1:3? This is confusing.

Author Response

Dr. Oisik Das                                               May, 23, 2020

Guest Editor, POLYMERS

Assistant Professor, Luleå University of Technology, Sweden

Dear Dr. Das:

In responding to the comments made by the reviewers on the paper entitled “Characterization, biocompatibility, and optimization of electrospun SF/PCL/CS composite nanofibers,” by Hua-Wei Chen and Min-Feng Lin, the following reply is made:

#REVEIWER 1:

  1. In the introduction, the authors didn’t elaborate clearly why they chose to make a composite fiber membrane with the selected components including silk fibroin, PCL and chitosan. It could be implied that these components have all been used individually for bio-related applications. However, the authors fail to tell the readers why they chose them and why as composites. There are already many nanofiber membranes for the proposed applications as described in this manuscript.

Reply: As indicated by the Reviewer, the reasons for the selected SF, PCL, and CS was supplemented in the part “Introduction” of the revised manuscript. Silk fibroin (SF), which is extracted mainly from silkworms, has various properties, including good biocompatibility, biodegradability, morphological flexibility, mechanical properties, a low inflammatory response, non-toxicity, and non-carcinogenicity, and it can promote cell adhesion, migration, and the proliferation of cell ligands [8, 20, 21]. However, the β-sheet secondary structure of pure SF seems to impede the electrospinning process, and the mechanical properties of neat SF electrospun fibers was poor [20]. Herein, Polycaprolactone (PCL), the most commonly used synthetic polymer, was chosen to blend with SF. The PCL is widely used in tissue engineering and drug delivery applications due to its good mechanical properties and biodegradability. PCL has limitations to its biological activity, hydrophobicity, and bacterial degradation, therefore the PCL cannot provide the adhesion environment required for cells [2]. PCL have renowned mechanical property and do not have ionizable side groups in their structures, such as –COOH and –NH2, which occur on the natural polymer chitosan (CS) and on several anionic polysaccharides and proteins, respective. The CS is an amino polysaccharide derived from chitin that has excellent biological properties, such as biocompatibility, biodegradability, hydrophilicity, non-toxicity, and antithrombotic and antimicrobial activity. Chitosan that possesses positively-charged groups (amine groups) is likely to interact with negatively-charged cell membranes via electrostatic interaction. Furthermore, the antimicrobial activity of chitosan effectively increases the permeability of negatively charged cell membranes to disrupt and release intracellular compounds [22]. Therefore, the SF and its association with other components (PCL and CS) are carried out to give better mechanical properties to the smooth nanofibers for the tissue engineering. Thus, SF/PCL/CS composite nanofiber scaffolds for the application of dressings were fabricated with PCL polymer as precursor using electrospinning techniques in this study.

  1. Novelty of the work is established. The objective is not coincided with the work presented in this study. In Table 1, is the relative ratio of chitosan and SF/PCL going to influence the results? Also, is it going to be easier for readers to follow if the ratios of each component were addressed as the percentages out of everything in total, e.g., in Level 1, chitosan takes 0.5 wt.%, SF takes 49.75 wt.%, and PCL takes 49.75 wt.% of the formulation? Or maybe the contents of chitosan are not as important from the following discussions in the manuscript since it takes only less than 1% for all the formulations.

Reply: As indicated by the Reviewer, the objective of this study has been improved in the revised manuscript. As the suggested by the Reviewer, the ratios of each component were addressed in Table for easily reading (e.g., in Level 1, chitosan takes 0.50 wt.%, SF takes 5.00 wt.%, and PCL takes 5.00 wt.% of the formulation). The SF and PCL were dissolved in formic acid to obtain 10 wt.% concentrations.

Table 1. Actors and levels used in the experiment

Ratio of silk fibroin and polycaprolactone

Chitosan content (%)

Flow rate (mL/h)

Voltage (kV)

Level 1

1:1 (5.00%:5.00%)

0.50

0.2

15

Level 2

1:2 (3.33%:6.67%)

0.75

0.3

20

Level 3

1:3 (2.50%:7.50%)

1.00

0.4

25

  1. There is a spelling error in Table 4, by blanck, the authors probably meant Also, since the ratio of silk fibroin and polycaprolactone has already been defined as factor A in Line 166-167, why don’t you make it simple and use the defined term in the following texts or not defining factor A at all since it appeared only once in the whole manuscript?

Reply: As mentioned by the Reviewer, the mistakes of Table 4 been corrected and the definition as factor A also been deleted in the revised manuscript.

  1. Starting from Line 177 to Line 179, the authors brought up the optimal parameters and conditions for fabrication. Although it is clear from previous discussion that factor A plays the most important role among all the four factors that were controlled, how each parameter in the three levels were determined was not explained. In other words, where does the conclusions about optimal parameters starting from Line 177 to Line 179 come from? Were they selected based on minimum S/N ratio? If they were selected based on the diameter of the nanofibers, under which circumstance, the smallest is believed to be the best, shouldn't the optimal ratio between SF and PLC 1:2 rather than 1:3? This is confusing.

Reply: As mentioned by the Reviewer, the typed errors have been rechecked and corrected in the part “Conclusion” of revised manuscript. The optimum combination for obtaining the smallest fiber diameter (170 nm) with a smooth and uniform distribution was determined to be a ratio of SF to PCL of 1:2, a flow rate of 0.3 ml/hr, and an applied voltage of 25 kV at a needle tip-to-collector distance of 15 cm (position). In a Taguchi designed experiments, higher values of the signal-to-noise ratio (S/N) identify control factor settings that minimize the effects of the noise factors. Using Equation (1) of the smaller the better method (i.e. the smallest diameter of the nanofibers was selected based on maximum S/N ratio), the S/N ratio of the fiber fineness (the diameter of the nanofibers) in the SEM micrograph of the nanofiber could be calculated. As the indicated by the Reviewer, the smallest is believed to be the best, and the confusing sentence have been amended and rewritten in the revised manuscript.

I really appreciate the careful devotion by the Editor and the Reviewers for such resourceful comments. If you have any further concerns about this paper, please feel free to contact me.

Sincerely Yours,

Hua-Wei Chen, Ph.D.,

Department of Chemical and Materials Engineering, National IlanUniversity,

No.1, Sec. 1, Shennong Rd., Yilan City, Yilan County 26047, Taiwan;

Tel.: 886-3-9317498, FAX: 886-3-9357025

E-mail: [email protected]; [email protected]

Reviewer 2 Report

The subject of the manuscript „Characterization, biocompatibility, and optimization of electrospun SF/PCL/CS composite nanofibers” by Hua-Wei Chen and Min-Feng Lin is in good relevance with the scope of Polymers. This study and its results demonstrate an interesting potential application of silk fibroin (SF), polycaprolactone (PCL) and chitosan (CS) composite nanofibers obtained by electrospinning as wound dressings. After analyzing the manuscript I have some suggestions and questions for the authors listed below:

  1. Check the manuscript as some typos occurred, e.g. Abstract line 12: Why antimicrobial activity is underlined? Some other underlined words appear also in the manuscript. This should be corrected. 
  2. Abstract: MTT test is not a cytotoxicity test? Why is mentioned separately?
  3. Abstract, Line 18: What cultures is about? Be more specific.
  4. Electrospinning of PCL/silk fibroin was done also by other researchers. The authors should compare their research with the similar ones and they should better highlight the novelty of their research.
  5. Figure 1 - (a) and (b) letters should be next to the figure not on a separate page.
  6. Table 2 should be moved at the 3.1.3. section.
  7. At cytotoxicity tests section is not described how the electrospun sample influence or affects cells morphology. I recommend to be added some discussion about this aspect.
  8. Exact values of measured mean diameter inhibition zone (antibacterial results section) should be given in the text.
  9. "gram-positive" I recommend be written as "Gram-positive".
  10. Figure 8: Scale bar should be more visible.
  11. Why the solution conductivity was not considered as an important parameter that influences the electrospinning?

Author Response

Dr. Oisik Das                                               May, 23, 2020

Guest Editor, POLYMERS

Assistant Professor, Luleå University of Technology, Sweden

Dear Dr. Das:

In responding to the comments made by the reviewers on the paper entitled “Characterization, biocompatibility, and optimization of electrospun SF/PCL/CS composite nanofibers,” by Hua-Wei Chen and Min-Feng Lin, the following reply is made:

#REVEIWER 2:

  1. Check the manuscript as some typos occurred, e.g. Abstract line 12: Why antimicrobial activity is underlined? Some other underlined words appear also in the manuscript. This should be corrected?

Reply: As indicated by the Reviewer, the typos have been rechecked and corrected in the revised manuscript.

  1. Abstract: MTT test is not a cytotoxicity test? Why is mentioned separately?

Reply: As mentioned by the Reviewer, MTT test is a cytotoxicity test. Thus, we deleted the “MTT test” in the part “Abstract” of the revised manuscript.

  1. Abstract, Line 18: What cultures is about? Be more specific.

Reply: As indicated by the Reviewer, cell culture is the process by which cells from human tissue are grown in an incubator under controlled conditions in order to provide sufficient material for testing. The definition of viability of these mouse fibroblast L929 cell cultures are presented in Equation (3). The Equation (3) was used to calculate the reduction in viability of these mouse fibroblast L929 cell cultures exposed to a tested sample (i.e. SF/PCL/CS under optimal conditions) in comparison to the cell culture viability of group b:

                                     (3)

where OD570e is the average OD of the respective groups that were in contact with different lots of the product and OD570b is the average OD of all wells of group b. All values were final ODs after subtraction of background absorbance. The sentences were supplemented in the revised manuscript.

  1. Electrospinning of PCL/silk fibroin was done also by other researchers. The authors should compare their research with the similar ones and they should better highlight the novelty of their research.

Reply: As suggested by the Reviewer, the discussions of comparison with their researches have been supplemented in the revised manuscript. The porosity of the nine samples (L9 (34)) of the SF/PCL/CS nanofibers in this study was more than 80% and was significantly higher than the pure PCL scaffolds, which had 70% porosity, as reported by [29]. The highly porous SF/PCL/CS nanofibers could provide an appropriate environment for initial cell growth (by their structural stability), accelerated degradation (by their large surface area), and sustained delivery of bioactive molecules (by their high porosity).

The EWC was increased with the increasing chitosan addition due to functional groups of –OH and –NH2. The equilibrium water content (EWC) of all the nanofibers (chitosan additions of 0.25, 0.50, 0.75, and 1.00%) was in the range of 500 to 950% and was significantly higher than the YY0148-2006 standard for medical dressings [39] and the hydrogels with the highly attractive for tissue engineering [40]. The copolymerization with zwitterionic comonomer leads hydrogels with high equilibrium water content (EWC), up to 700% while maintaining mechanical robustness [40].

In cultures exposed to blank, negative control, positive control, and SF/PCL/CS, the viability was 100, 100, 13, and 57% at 24 hours of continued growth in the culture, respectively (Table 4). A tested product (SF/PCL/CS) has non-cytotoxic potential for application to tissue engineering when the cell culture viability increases to >50% in comparison to positive control (dimethylsulfoxide, DMSO), which was set at 13% viability. The study also proposed the similar results that the tested product could be considered non-toxic as the viability of these cultures exceeded 50% [30]. Furthermore, the viability of 57% for the tested product (SF/PCL/CS) in this study was higher than that of 50.2% for the CNTs-doped PLGA nanofibers [31]. The research [31] show that the PLGA, and the HNTs- or CNTs-doped PLGA nanofibers display appreciable MTT formazan dye sorption, corresponding to 35.6–50.2% deviation from the real cell viability assay data.

  1. Figure 1 - (a) and (b) letters should be next to the figure not on a separate page.

Reply: As indicated by the Reviewer, Figures have been separated as shown in Figure1 and Figure2.

  1. Table 2 should be moved at the 3.1.3. Section.

Reply: As indicated by the Reviewer, the result of “WVTR” in Table 2 and that of “Cytotoxicity tests” in Table 4 and was discussed at the 3.1.2. Section  and the 3.1.3. Section.

  1. At cytotoxicity tests section is not described how the electrospun sample influence or affects cells morphology. I recommend to be added some discussion about this aspect.

Reply: As suggested by the Reviewer, From Figure 3, DMSO treatment substantially altered the morphology and attachment of cells in concurrence with a significant reduction in cell viability. The results showed that the electrospun scaffolds (SF/PCL/CS) could support the attachment and the proliferation of mouse fibroblast L929 cells. In addition, the cells cultured on the scaffolds exhibited normal cell shapes. The obtained results confirmed the potential for use of the electrospun fiber as scaffolds for skin tissue engineering. The sentences have been supplemented in the revised manuscript.

  1. Exact values of measured mean diameter inhibition zone (antibacterial results section) should be given in the text.

Reply: As indicated by the Reviewer, the results (Table 5) revealed the mean diameter of inhibition zone for E. coli with the chitosan amounts of 0.25, 0.5, 0.75, and 1.00% were 65, 63, 64, and 63 mm, proving the strong antibacterial property. Moreover, the samples with the chitosan amounts of 0.25, 0.5, 0.75, and 1.00% exhibited the inhibition zone diameters of 58, 55, 51, and 55 mm S. aureus, suggesting the antibacterial property was determined by the chitosan amounts.

Table 5. Effect of chitosan addition on the mean diameter of inhibition zone of E. coli and Saureus at different chitosan amounts

Name of the sample

Conc.(%)

Mean diameter of inhibition zone (mm)

Test organisms

Staphylococcus aureus

Escherichia coli

Chitosan

0.25

58±4

65±0

0.50

55±1

63±4

0.75

51±1

64±0

1.00

55±1

63±2

  1. "gram-positive" I recommend be written as "Gram-positive".

Reply: As recommend by the Reviewer, the mistake was corrected in the revised manuscript.

  1. Figure 8: Scale bar should be more visible.

Reply: As indicated by the Reviewer, the resolution of the scale bar in Figure 8 was improved in the revised manuscript.

  1. Why the solution conductivity was not considered as an important parameter that influences the electrospinning?

Reply: As recommend by the Reviewer, we fully agree the REVIEWER’s comments that the effect of the solution conductivity should be considered because the solution conductivity plays an important role in the electrospinning. However, it is difficult to carry on due to the limitation of instruments. In the future study, we will attempt to investigate the effect of the solution conductivity.

I really appreciate the careful devotion by the Editor and the Reviewers for such resourceful comments. If you have any further concerns about this paper, please feel free to contact me.

Sincerely Yours,

Hua-Wei Chen, Ph.D.,

Department of Chemical and Materials Engineering, National IlanUniversity,

No.1, Sec. 1, Shennong Rd., Yilan City, Yilan County 26047, Taiwan;

Tel.: 886-3-9317498, FAX: 886-3-9357025

E-mail: [email protected]; [email protected]

Round 2

Reviewer 1 Report

The modified manuscript has been improved.

Author Response

#REVEIWER 1:

  1. Review Report Form:English language and style are fine/minor spell check required 

Reply: As noted by the Reviewer, English language and style have been rechecked and corrected in the revised manuscript. 
